# The Global Spread Pattern of Rat Lungworm Based on Mitochondrial Genetics

**DOI:** 10.3390/pathogens12060788

**Published:** 2023-05-31

**Authors:** Xia Tian, Shen Chen, Lei Duan, Yingjun Qian, Hongmei Li, Shan Lv

**Affiliations:** 1National Institute of Parasitic Diseases, Chinese Center for Disease Control and Prevention, Shanghai 200025, China; tx09871@163.com (X.T.); nematode@163.com (S.C.); duanlei@nipd.chinacdc.cn (L.D.); qianyj@nipd.chinacdc.cn (Y.Q.); lihm@nipd.chinacdc.cn (H.L.); 2Key Laboratory on Parasite and Vector Biology, National Health Commission, Shanghai 200025, China; 3WHO Collaborating Centre for Tropical Diseases, Shanghai 200025, China; 4National Center for International Research on Tropical Diseases, Ministry of Science and Technology, Shanghai 200025, China; 5Chinese Center for Tropical Diseases Research, Shanghai 200025, China

**Keywords:** *Angiostrongylus cantonensis*, *Angiostrongylus mackerrasae*, *Angiostrongylus malaysiensis*, rat lungworm, mitochondrial gene, distribution, phylogeny

## Abstract

Eosinophilic meningitis due to rat lungworm, *Angiostrongylus cantonensis*, is a global public health concern. Human cases and outbreaks have occurred in the new endemic areas, including South America and Spain. The growing genetic data of *A. cantonensis* provides a unique opportunity to explore the global spread pattern of the parasite. Eight more mitochondrial (mt) genomes were sequenced by the present study. The phylogeny of *A. cantonensis* by Bayesian inference showed six clades (I–VI) determined by network analysis. A total of 554 mt genomes or fragments, which represented 1472 specimens of rat lungworms globally, were used in the present study. We characterized the gene types by mapping a variety of mt gene fragments to the known complete mt genomes. Six more clades (I2, II2, III2, V2, VII and VIII) were determined by network analysis in the phylogenies of *cox*1 and *cyt*b genes. The global distribution of gene types was visualized. It was found that the haplotype diversity of *A. cantonensis* in Southeast and East Asia was significantly higher than that in other regions. The majority (78/81) of samples beyond Southeast and East Asia belongs to Clade II. The new world showed a higher diversity of Clade II in contrast with the Pacific. We speculate that rat lungworm was introduced from Southeast Asia rather than the Pacific. Therefore, systematic research should be conducted on rat lungworm at a global level in order to reveal the scenarios of spread.

## 1. Introduction

*Angiostrongylus cantonensis* is the major cause of human neural angiostrongyliasis [1]. Infection with this nematode often results in eosinophilic meningitis (EM) and other disorders of the central nervous system [2]. Humans are vulnerable to infection because of the varied contact opportunities with mollusk intermediate hosts and paratenic hosts. EM occurs following ingestion of undercooked or raw snails, slugs, freshwater shrimp or crabs, lizards, and even contaminated vegetable salad with mollusk slime [3].

*A. cantonensis* was first discovered from the pulmonary arteries and heart of rats in southern China in 1933 [4]. Human EM due to *A. cantonensis* was first reported in 1945 [5], followed by EM outbreaks in Pohnpei, of the Caroline Islands of the Pacific between 1947 and 1948 [6]. However, the health concern was not noted in the early 1960s when an outbreak occurred in Tahiti [7]. The subsequent surveys indicated that EM due to *A. cantoensis* was endemic in Southeast Asia and the Pacific [8]. Cases reported in recent years indicate that *A. cantoensis* constitutes an emerging zoonosis worldwide [9,10,11].

The expansion of the endemic range of *A. cantonensis* has been, to an extent, attributed to biological invasion. The common definitive host, the black rat *Rattus rattus* and the Norway rat *R. norvegicus*, are listed as the world’s top invasive species [12]. The growing shipping industry, particularly for cargo, drove the global spread of the two Rattus species and thus *A. cantonensis*. The intermediate host snail, *Achatina fulica*, was thought to play a key role in the spread of *A. cantonensis* in World War II, particularly in the Pacific islands [13]. Indeed, the land snail occupied the islands quickly in the 1940s and 1950s [14]; however, some human cases attributed to the consumption of *A. fulica* were reported earlier [15]. The snail has also recently begun to be considered an important factor in the occurrence of *A. cantonensis* in South America [16]. Another global invasive snail species is *Pomacea canaliculata*, which has been widely distributed in Southeast and East Asia since the 1980s [17]. The freshwater snail has driven the emergence of human neural angiostrongyliasis in the region [18]. The continuous invasion and re-introduction of definitive and intermediate hosts make the global spread of *A. cantonensis* more complex.

Rat lungworm commonly refers to *A. cantonensis*. However, rat lungworm also includes *A. mackerrasae* and *A. malaysiensis*, which were not distinguished from *A. cantonensis* until the late 1960s [19,20]. Although they have a nearly consistent life cycle, their geographical range and preference for a definitive host are different. In contrast with the global distribution of *A. cantonensis*, *A. mackerrasae* is confined to the east coast of Australia [21] and *A. malaysiensis* is mainly distributed in Thailand, Malaysia and Indonesia [19,22]. The most common definitive host of *A. mackerrasae* is the bush rat *R. fuscipes* and swamp rat *Rattus lutreolus* [21], while *A. malaysiensis* prefers the Malaysian field rat *R. tiomanicus* and the ricefield rat *R. argentiventer* [23]. Nevertheless, the two species have also been discovered from the global invasive *R. norvegicus*. Therefore, it is possible that the two species of *Angiostrongylus* may be distributed beyond the original regions. In addition, the diagnosis of most infections in human and animal was based on larval morphology and immunology, the infections due to *A. mackerrasae* and *A. malaysiensis* were probably misdiagnosed. For example, a recent report has shown through molecular evidence that *A. malaysiensis* can infect humans [24].

The genetic data of rat lungworms are increasingly prevalent around the world, providing an opportunity to reveal the global spread pattern of *A. cantoensis* and check the possibility of a range expansion for *A. mackerrasae* and *A. malaysiensis*. However, the molecular markers are diverse, with a focus on mitochondrial (mt) genes, including cytochrome c oxidase 1 gene (*cox*1) [25,26], cytochrome b gene (*cyt*b) [27], NADH dehydrogenase 1 gene (*nad*1) [28], ribosomal subunit RNA gene [29], and even complete mt genomes [30]. In the present study, we will characterize the available complete mt genomes and map the known gene fragments to the mt genomes so that we can categorize the samples from multiple published works. Finally, the global distribution patterns of the gene types will be revealed and the global spread pattern of *A. cantoensis* will be inferred.

## 2. Materials and Methods

### 2.1. Mitochondrial Genomes

The present study will provide eight full mt genomes of *A. cantonensis*. The adult worms of *A. cantonensis* were collected from the pulmonary arteries of wild rats during the first national survey in China [31], or from Sprague–Dawley rats that were infected by third stage larvae from snails or slugs. Nine female worms from seven collecting sites were used for sequencing mt genomes (Table 1). One of the full sequences had been published (GQ398121). The other eight were characterized by the same methods and used for the present study [32]. In addition, the mt genomes of *A. cantonensis*, *A. malaysiensis* and *A. mackerrasae* that are deposited in GenBank were also included in the present analysis.

### 2.2. Mitochondrial Gene Sequences

Since our analysis strategy is to map the mt genes to the known complete mt genomes and then determine the types of each sequence, all mt gene sequences of *A. cantonensis*, *A. malaysiensis* and *A. mackerrasae* were collected from GenBank and used for the present study. Meanwhile, the collection information of specimens, e.g., location, year, species, gene name, sample size were also collected. For the sequences which had been published in papers but not yet submitted to GenBank, we collected the sequences directly from the papers including Appendix A.

### 2.3. Phylogenetic Analysis

The complete mt genomes of *A. cantonensis*, *A. malaysiensis* and *A. mackerrasae* were used to construct phylogeny with the *A. vasorum* (JX268542) and *A. costaricensis* (GQ398122) as outgroups. A phylogenetic tree was constructed under Bayesian inference, performed in MrBayes version 3.2.7 [31]. Prior to Bayesian inference, the best fit nucleotide substitution model (GTR+G) for the dataset was determined using a hierarchical likelihood ratio test in jModeltest version 2 [32]. The posterior probabilities were calculated using Markov chain Monte Carlo (MCMC) simulations. At the end of this run, the average standard deviation of split frequencies was below 0.01, and the potential scale reduction factor was reasonably close to 1.0 for all parameters. A consensus tree was visualized and edited by Mesquite version 3.70 [36] or FigTree version 1.4.4 (http://tree.bio.ed.ac.uk/software/Figtree/; accessed on 28 May 2023).

The group of complete mt genomes was also submitted to network analysis by TCS version 1.21 [36]. Subgroups were determined with a connection limit of 1% or 132 steps and considered as the clades in the phylogenetic tree mentioned above.

The mt gene sequences downloaded from GenBank and papers were first categorized by the gene name. The sequences of the same gene were mapped to the complete mt genomes by alignment in clustal X. The sequences might be divided into subgroups when the overlapped length was less than 50%. After being trimmed as necessary, the grouped sequences were submitted to DnaSP version 5 [37] and the haplotypes were determined. The phylogenetic trees based on the haplotypes were inferred following the guide described above.

Some studies characterized the sequence of different mt genes of the same specimen, e.g., *cox*1 and *cyt*b. Only one gene for a single specimen was taken into analysis. The gene *cox*1 was of priority because it was used broadly, followed by *cyt*b and small ribosomal RNA gene. Large ribosomal RNA gene and nad1 were excluded because *cox*1 and small ribosomal RNA gene from the same specimens were included.

The gene fragments deduced from the 18 whole mt sequences of *A. cantonensis*, *A. malaysiensis* and *A. malaysiensis* were included as index markers in the individual gene phylogenetic trees. Each haplotype was assigned to a specific clade as delineated by whole mt genomes according to the index markers. New clades could be defined if the haplotypes in the clades were separated by network analysis with a connection limit of 1%, the same threshold as in analysis of full mt genomes. If a new clade was clustered with a specific known clade inferred according to the full mt genomes, the new clade would have the same name with a suffix of 2. Otherwise, a new name would be given to the new clade.

### 2.4. Mapping the Distribution Pattern

The coordinates of specimen collecting sites were either directly collected from the published papers or determined by Google Earth by inputting the location names which were extracted from the papers and GenBank. Each gene sequence from known locations was assigned to a specific clade determined in phylogenetic analysis.

The geographic distributions of gene types were mapped and visualized in a geographical information system (ArcGIS version 10.1). We merged some collecting sites for better display when the distance among them was less than 0.8 degrees at the global level and less than 0.5 degrees at the subregion level.

## 3. Results

A total of 18 mt genomes of rat lungworms were used in the analysis, including 10 from GenBank and 8 from the present study (Table 1, File S1). The genetic distance among three species of rat lungworm was around 0.12, while the genetic distance among various strains of *A. cantonensis* ranged from 0.001 to 0.056 (Table 2). The phylogenetic and network analyses showed that 15 mt genomes of *A. cantonensis* collapsed into six clades, encoded as clade I to clade VI (Figure 1). One genome (KT186242), previously identified as *A. cantonensis,* had closer genetic relation to *A. malaysiensis* (KT947979) according to the present study. Another genome (MN793157) belonged to *A. mackerrasae*.

The individual gene phylogenetic trees of *cox*1, *nad*1 and small ribosomal RNA genes from 18 complete mt genomes were re-constructed by Bayesian inference (Appendix A). When incorporated into the individual gene phylogenetic trees, the gene fragments deduced from the whole mt sequences all remained in their original clades, as expected.

A total of 554 mt genomes or fragments, which represented 1472 specimens of rat lungworms globally (Table 3), were used in the present study. The specimens were collected from 224 sites. According to our analysis one mt genome (KT186242), 13 *cox*1 sequences and 64 *cyt*b sequences were *A. malaysiensis* and had previously been identified as *A. cantonensis*.

There are 327 *cox*1 sequences, including 160 fragments from GenBank, 18 mt genomes and 149 sequences from our previous study. One sequence (LC515249) was excluded due to the lack of overlap with any other sequence. The rest of the 326 sequences were divided into two subgroups according to the criteria of an overlap of less than 50%. The first subgroups included 308 sequences, which belonged to 112 haplotypes. All six of the clades (I ~ VI) based on the complete mt genomes were observed in the phylogenetic tree based on the first subgroups of *cox*1 genes. Meanwhile, another four clades were identified and named as Clade I-2, Clade II-2, Clade III-2 and Clade V-2 based on their relations to the six known clades (Figure 2). The second subgroup included 18 sequences, belonging to 11 haplotypes. All the sequences fell into Clade II (Figure 3).

Ten *cyt*b sequences from the same specimens of *cox*1 were excluded in the analysis. The rest of the 195 *cyt*b sequences, including 177 from GenBank and 18 from complete mt genomes, collapsed into 67 haplotypes. Eight clades were found in the phylogenetic analysis (Figure 4), including six that were determined according to complete mt genomes and two distinctive clades named Clade VII and VIII. The average genetic distance of sequences in Clade VIII to the other clades of *A. cantonensis* was over 5% (Appendix A). The difference could be as high as 7%.

Twelve sequences of small and large ribosomal RNA genes were from the same specimens. Taking into consideration the smaller ribosomal subunit gene (ON747257), 13 small ribosomal RNA genes were used in the analysis. The sequences formed five haplotypes. According to network analysis, three haplotypes belonged to Clade II (Figure 5), while another two haplotypes fell into Clade I and III, respectively.

The gene types (clades) based on phylogenetic and network analyses were mapped in the world (Figure 6, Appendix A). A much higher diversity of clades was observed in Southeast and East Asia. In contrast, almost all 81 specimens of *A. cantonensis* beyond the region belonged to Clade II, except for two (Clade IV) from Hawaii and one (Clade V) from Rio de Janeiro. In Asia, the Annamite range might be considered the genetic barrier for *A. cantonensis* (Figure 7). The gene types of Clade III, IV, V, V-2, VI and VIII were mainly distributed east of the Annamite range. On the contrary, Clade I, III-2 and VII were commonly found to the west of the Annamite range. *A. malaysiensis* was commonly found in in the west of Annamite range. However, it was also discovered in Taiwan according to the present analysis. *A. mackerrasae* was only observed in Australia.

The *cox*1 sequence types in Clade II were further categorized into ten subgroups (Figure 8). According to the geographic distribution of the subgroups, the haplotype diversity beyond Southeast and East Asia was 0.944, which was higher than that of 0.4641 in Asia (Figure 9). Clade II was not commonly observed in Asia according to available data. For example, Clade II was not discovered in the mainland of China and the northern part of the Greater Mekong subregion, though intensive sampling was undertaken and the high genetic diversity of *A. cantonensis* was observed in the region. Another feature of clade distribution in Asia was that none of the seven clades, except for Clade II-G, was simultaneously observed in two or more sites.

Clade II-G is the most common type, accounting for 48.57% of Clade II and found in seven sites in the world, five of which were located in the Pacific islands. Meanwhile the clade was also extended to the southeast shore area of Brazil. In contrast with the Pacific, the new world shows a higher diversity of clades, including the predominant clades II-D and II-E. Although the clade II-D1 and clade II-D3 were observed on Taiwan island and eastern Australia, 90% of the samples of Clade II-D were from the new world and Spain, while Australia shared two clades with Southeast Asia and South America.

## 4. Discussion

A total of 18 complete mt genomes of rat lungworms are currently available. Only one mt genome is *A. mackerrasae* and *A. malaysiensis*, respectively. The genetic distance among the species is over 0.11, while that among the different geographical strains within species is less than 0.06. Therefore, the species of rat lungworm could be definitively distinguished based on genetic distance. The genetic distance of the mt genome (KT186242), previously identified as *A. cantonensis*, was over 0.11 to any known strain of *A. cantonensis*. In contrast, the genome is almost identical to the mt genome (KT947979) diagnosed as *A. malaysiensis*. Therefore, KT186242 should be *A. malaysiensis*.

There were two hypotheses about the origin of *A. cantonensis*. Earlier opinion indicated that the parasite originated from Africa, which was supported by the fact that the discovery of *A. cantonensis* coincided with the spread of the African land snail *A. fulica* to Southeast Asia [38]. However, the theory of Asian origin later became popular. It was thought that the parasite was originally endemic to Southeast Asia and spread by the shipping rats, *R. rattus* and *R. norvegicus*, due to extensive traveling [13]. Our results show a much higher genetic diversity of *A. cantonensis* in Southeast and East Asia, supporting the latter theory.

Although intensive sampling occurred in China, Japan and the Great Mekong subregion, the common gene types (e.g., Clade II-D and Clade II-E shown in Figure 9) in America were rarely found in the region mentioned above. It was therefore believed that the gene types must be from anywhere else in Southeast Asia. Of note, Clade II was genetically close to Clade I. The latter was found only in Thailand and Myanmar, where *A. malaysiensis* co-occurred. The phenomenon of significant correlation between genetic and geographical distances is commonly observed in population genetics [39]. Therefore, Clade II was possibly from either Myanmar or the lower reach of the Mekong River. Similarly, Clade V showed longer genetic distance from any other clade within *A. cantonensis*, though the clade co-occurred with clade IV in China and Japan. We did not obtain any genetic evidence from the Philippines, but the country holds the potential to harbor the Clade V as a local gene type. A total of 124 *cox*1 samples from 24 collecting sites were available in Japan, and they consisted of 4 clades. However, the haplotype diversity of the samples was very low compared with Southeast Asia and the south of China [28,40]. Only seven haplotypes were identified, including three unique haplotypes within a single sample. Therefore, *A. cantonensis* in Japan might have been introduced from Southeast Asia and/or the south of China.

Our results show that there might be a genetic barrier between the Greater Mekong subregion and the south of China. The geographical isolation due to the Annamite range located in Vietnam may play a key role in the genetic divergence of *A. cantonensis*. Previous studies have indicated different evolutionary trajectories on western and eastern sides of the Annamite Mountain range [41]. The case of liver fluke also supports this opinion. Two species of liver fluke, i.e., *Clonorchis sinensis* and *Opisthorchis viverrini*, are widely distributed in the region. The former is distributed northeast of the Annamite range, while the latter is endemic in the southwest [42,43].

Clade II is the overwhelming gene type beyond Southeast and East Asia except for a small number of samples of Clade IV and Clade V in Hawaii and Rio de Janeiro, respectively. The conclusive global spread route of rat lungworm could not be established based on the present available genetic data. However, our findings imply that the gene types on the Pacific islands, where EM outbreaks occurred between the 1940s and 1960s, show identical and low diversity, which implies that its re-introduction after the Pacific War is less plausible [13]. Australia is the only country where *A. cantonensis* was reported earlier [21]. Our findings show there was no relation between Australia and the Pacific based on the available evidence, while the majority of samples are genetically close to those from Thailand. The new world, recently identified as endemic areas, probably showed a distinct haplotype structure from the Pacific and hence might have been directly introduced to the rat lungworm from Southeast Asia.

Compared with the global distribution of *A. cantonensis*, *A. mackerrasae* and *A. malaysiensis* are endemic locally. According to our results *A. mackerrasae* was only found on the eastern shore of Australia. Although *A. mackerrasae* was also discovered from *R. norvegicus*, it shows a higher susceptibility to local Australian rodents, e.g., *R. fuscipes*, *R. lutreolus* and *Melomys cervinipes* [21]. Therefore, *A. mackerrasae* was not endemic beyond the original region. *A. malaysiensis* showed a wider range of definitive hosts than *A. mackerrasae*. Although *A. malaysiensis* seems more susceptible to *R. tiomanicus* and *R. argentiventer*, the parasite shared 80% of definitive hosts with *A. cantonensis*, including the most invasive *R. norvegicus* and *R. rattus* (unpublished data). Our present study indicates that *A. malaysiensis* had gone beyond the original region and established local transmission in eastern Taiwan, since the worms had been isolated from the intermediate host *A. fulica* [44,45].

In order to reveal the complex spread pattern of *A. cantonensis*, we suggest the following research priorities. First, more mt genomes of rat lungworm should be characterized. Our present study identified new clades with complete mt genome sequences not yet available. Second, we need to fill the gap of genetic information in the Philippines and Indonesia, and even Myanmar. Since these countries are neighboring to the Greater Mekong subregion and East Asia, the mt genetic information of rat lungworms will be invaluable in constructing the phylogeny. Unfortunately, the large-scale survey has not been conducted and the mt genetic information has been lacking to date, though rat lungworms have been reported in the Philippines and Indonesia since the 1950s. Third, systemic sampling around the world is proposed to update either rare specimens or a small number of collecting sites beyond the Greater Mekong subregion and East Asia. In addition, the types of host animals, i.e., mollusks and rodents, were different in previous studies. It should be noted that the host preferences of species or strains of rat lungworm may cause a bias, and hence lead to false conclusions.

## 5. Conclusions

We showed an integrated global distribution of gene types by mapping various fragments to the known complete mt genomes. The new endemic areas including the new world and Spain showed different compositions of gene types to Southeast and East Asia and even the Pacific. Therefore, we need to conduct systematic research on rat lungworm at a global level in order to reveal the scenarios of spread.

## Figures and Tables

**Figure 1 pathogens-12-00788-f001:**
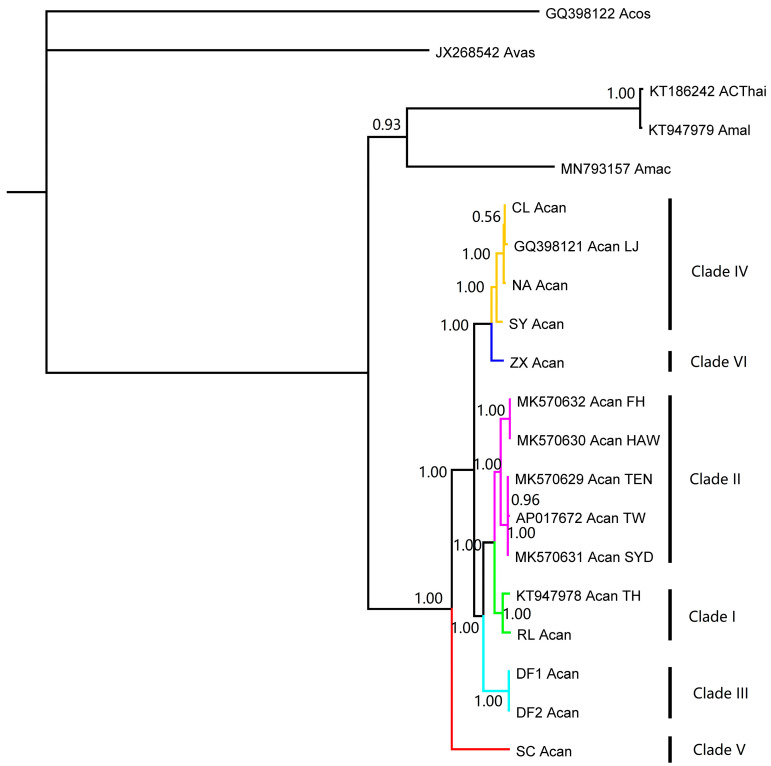
The phylogeny of rat lungworm inferred from mitochondrial genomes, with *A. vasorum* (JX268542) and *A. costaricensis* (GQ398122) as outgroups.

**Figure 2 pathogens-12-00788-f002:**
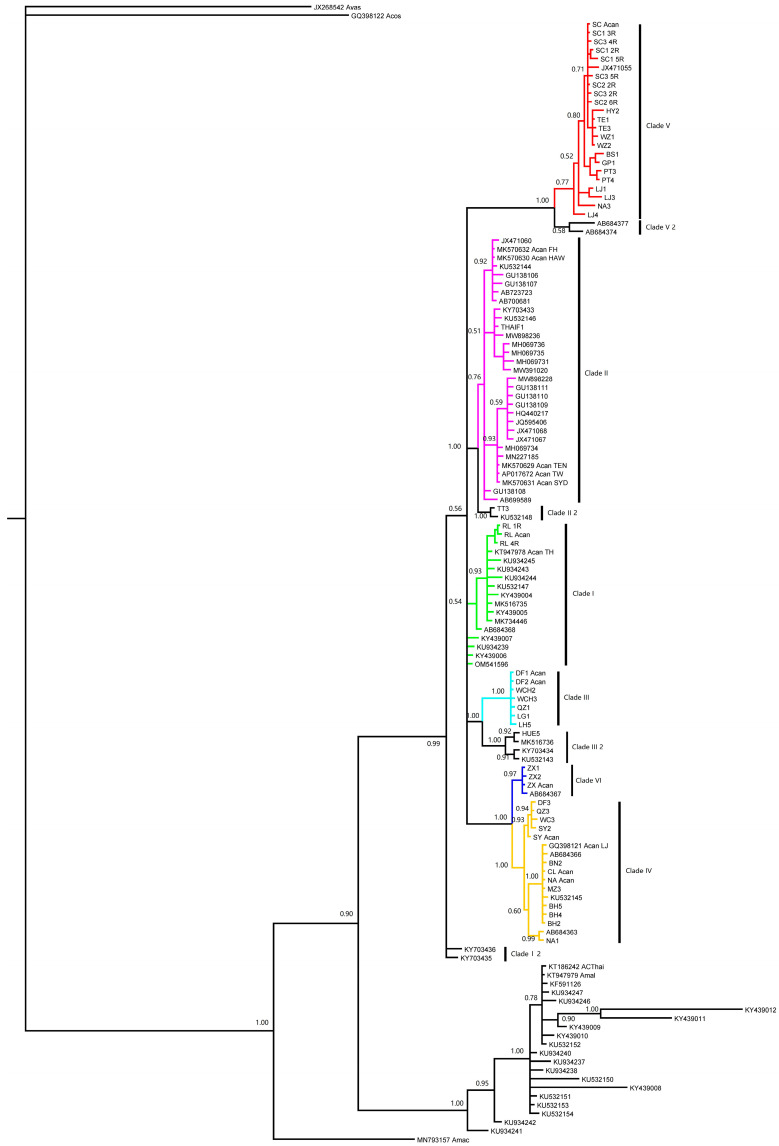
The phylogeny of rat lungworm inferred via the first group of *cox*1 haplotypes. The *cox*1 fragments of eighteen available full mitochondrial genomes are included as index of clades. The code and branch color is consistent with Figure 1. Four new clades, i.e., Clade I-2, Clade II-2, Clade III-2, and Clade V-2, are identified based on the network analysis.

**Figure 3 pathogens-12-00788-f003:**
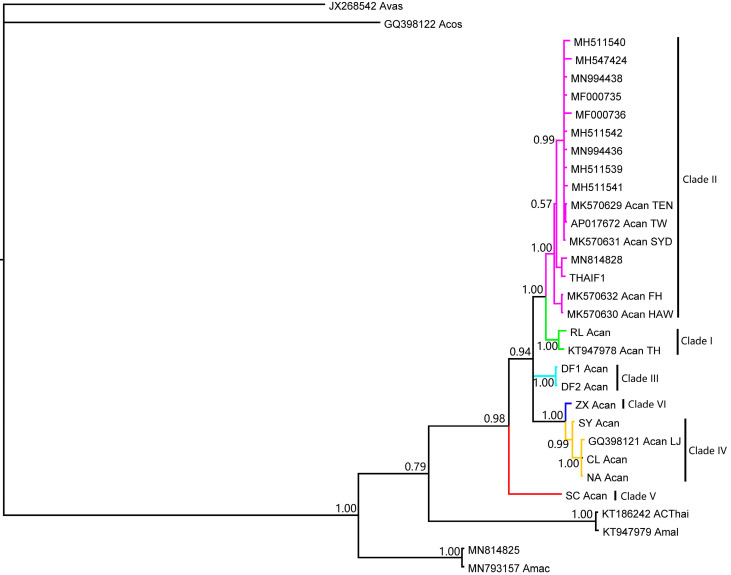
The phylogeny of rat lungworm inferred via the second group of *cox*1 haplotypes. The gene sequences used in this phylogeny showed <50% overlap with the sequences used in Figure 2.

**Figure 4 pathogens-12-00788-f004:**
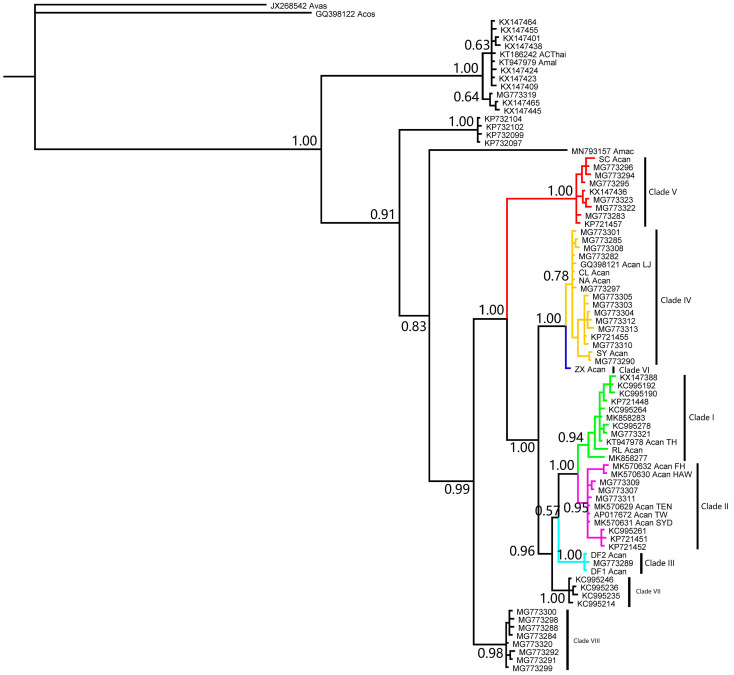
The phylogeny of rat lungworm inferred via the *cyt*b haplotypes. Two new clades, i.e., Clade VII and Clade VIII, were identified in the phylogeny compared with Figure 1.

**Figure 5 pathogens-12-00788-f005:**
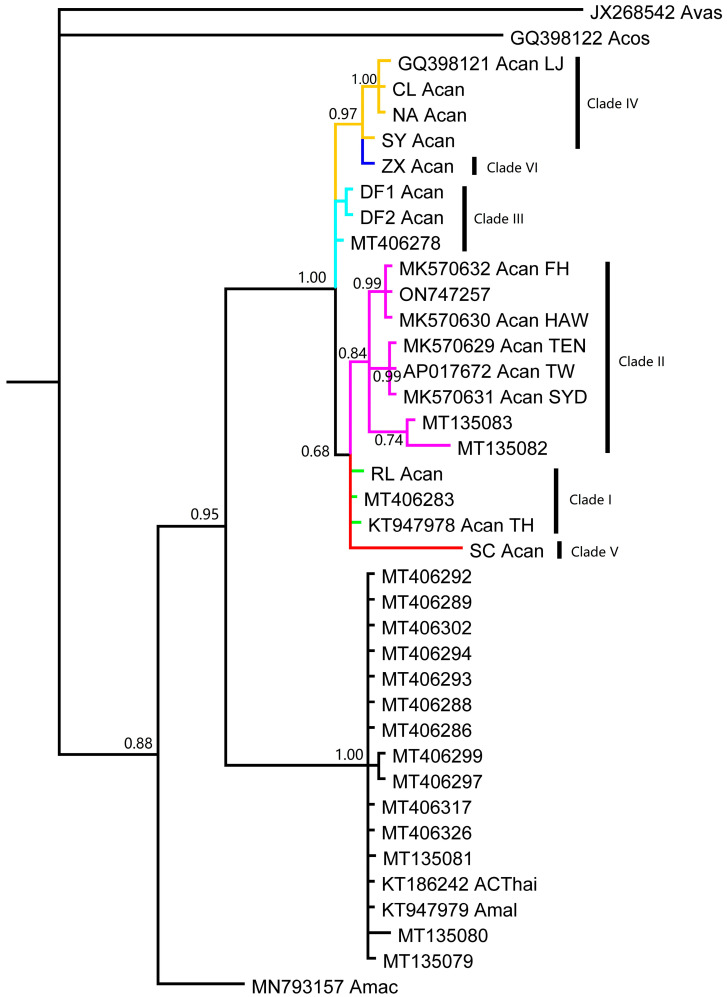
The phylogeny of rat lungworm inferred via the mitochondrial small ribosomal RNA gene.

**Figure 6 pathogens-12-00788-f006:**
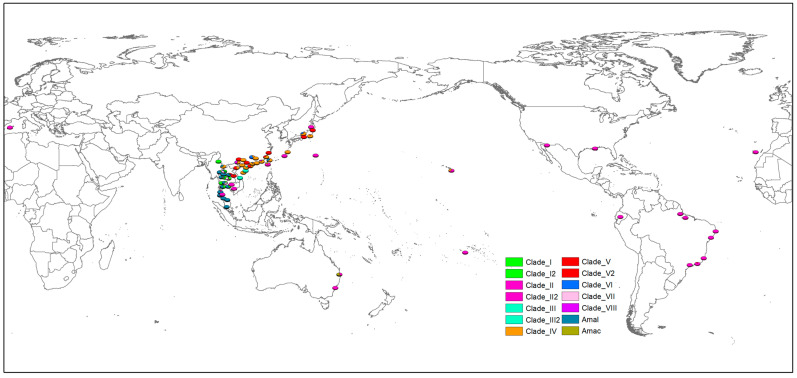
The global distribution of gene types of rat lungworm. The sites with a distance less than 0.8 degrees were combined and displayed at the geometric center. Amal: *A. malaysiensis*, Amac: *A. mackerrasae*.

**Figure 7 pathogens-12-00788-f007:**
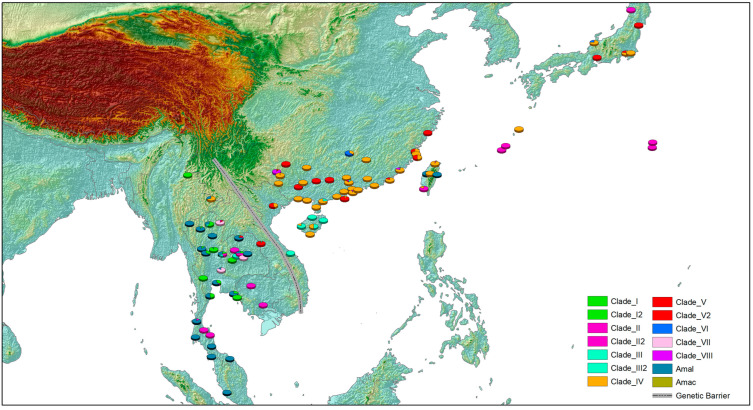
The distribution of gene types of rat lungworm in Southeast and East Asia. The sites with a distance of less than 0.5 degrees were combined and displayed at the geometric center. Amal: *A. malaysiensis*, Amac: *A. mackerrasae*.

**Figure 8 pathogens-12-00788-f008:**
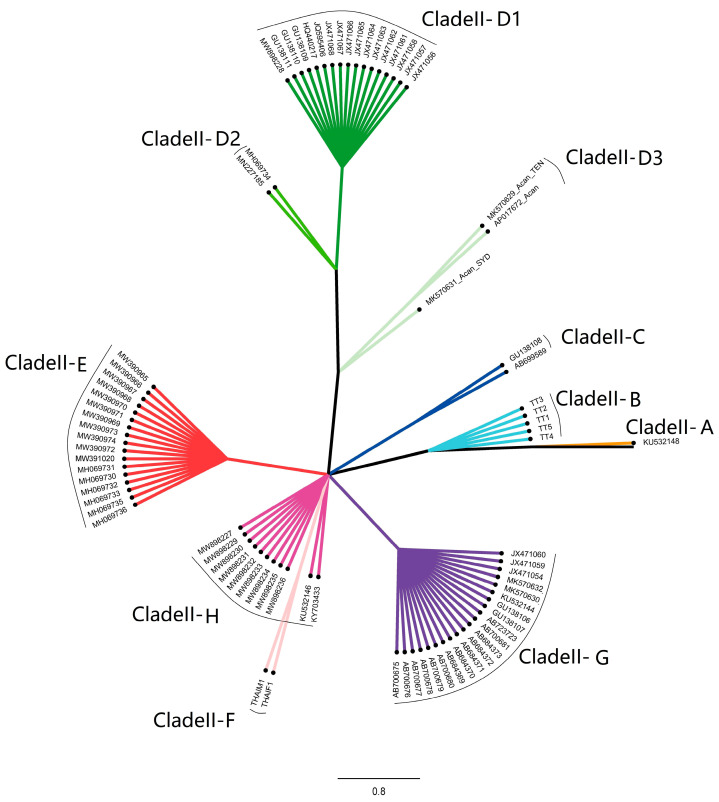
The phylogenetic relation of Clade II (*cox*1) subgroups.

**Figure 9 pathogens-12-00788-f009:**
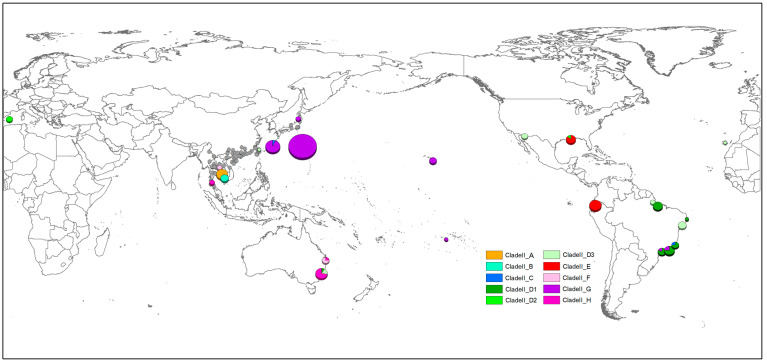
The global distribution of Clade II (*cox*1) with sample size. Grey dot indicate the distribution of other clades.

**Table 1 pathogens-12-00788-t001:** The mitochondrial genomes of *Angiostrongylus* used in phylogenetic analysis.

Species	Access Number	Location	Altitude	Longitude	References
*A. cantonensis*	AP017672	Taipei, China	25.0329	121.5655	[30]
	CL_Acan	Changle, China	25.9313	119.6288	this study
	DF1_Acan	Dongfang, China	19.0535	108.6521	this study
	DF2_Acan	Dongfang, China	19.0535	108.6521	this study
	GQ398121	Lianjiang, China	26.2052	119.5212	[32]
	KT947978	Thailand			[26]
	MK570629	Tenerife, Spain	28.2916	−16.6291	[30]
	MK570630	Hawaii, USA	19.6310	−156.0072	[30]
	MK570631	Mosman, Australia	−33.8293	151.2442	[30]
	MK570632	Fatu Hiva, French Polynesia	−17.6427	−149.4347	[30]
	NA_Acan	Nanao, China	23.4533	117.0971	this study
	RL_Acan	Ruili, China	23.9477	97.7854	this study
	SC_Acan	Shangchuan, China	21.6613	112.8016	this study
	SY_Acan	Sanya, China	18.3333	109.4333	this study
	ZX_Acan	Zixing, China	26.0365	113.2463	this study
*A. mackerrasae*	MN793157	Brisbane, Australia	−27.4620	153.0203	[33]
*A. malaysiensis*	KT186242	Thailand			[34]
	KT947979	Kuala Lumpur, Malaysia	3.1201	101.6545	[26]
*A. vasorum*	JX268542	Australia			[35]
*A. costaricensis*	GQ398122	Brazil			[32]

**Table 2 pathogens-12-00788-t002:** Genetic distance based on complete mitochondrial genomes.

	MN793157	KT186242	KT947979	CL_Acan	NA_Acan	GQ398121	SY_Acan	ZX_Acan	MK570632	MK570630	MK570629	AP017672	MK570631	KT947978	RL_Acan	DF1_Acan	DF2_Acan
KT186242	0.133																
KT947979	0.133	0.002															
CL_Acan	0.116	0.124	0.124														
NA_Acan	0.116	0.125	0.124	0.001													
GQ398121	0.117	0.125	0.125	0.002	0.002												
SY_Acan	0.115	0.123	0.123	0.008	0.007	0.009											
ZX_Acan	0.116	0.123	0.123	0.015	0.015	0.016	0.014										
MK570632	0.118	0.125	0.125	0.035	0.035	0.036	0.034	0.034									
MK570630	0.118	0.124	0.124	0.034	0.034	0.035	0.033	0.033	0.001								
MK570629	0.118	0.123	0.123	0.034	0.034	0.035	0.033	0.034	0.011	0.010							
AP017672	0.118	0.123	0.123	0.034	0.034	0.035	0.033	0.034	0.011	0.010	0.001						
MK570631	0.118	0.123	0.123	0.034	0.034	0.035	0.033	0.033	0.011	0.010	0.001	0.001					
KT947978	0.118	0.124	0.124	0.034	0.034	0.035	0.033	0.033	0.018	0.017	0.016	0.016	0.016				
RL_Acan	0.119	0.125	0.125	0.034	0.034	0.035	0.033	0.033	0.019	0.019	0.017	0.017	0.017	0.009			
DF1_Acan	0.118	0.124	0.124	0.035	0.035	0.036	0.033	0.034	0.030	0.029	0.029	0.029	0.029	0.029	0.030		
DF2_Acan	0.117	0.124	0.124	0.035	0.035	0.036	0.033	0.034	0.030	0.029	0.029	0.029	0.029	0.028	0.030	0.001	
SC_Acan	0.117	0.126	0.126	0.051	0.052	0.052	0.051	0.053	0.056	0.056	0.055	0.056	0.055	0.055	0.055	0.056	0.056

**Table 3 pathogens-12-00788-t003:** Mitochondrial gene sequences used in the present study.

Types	*A. cantonensis*	*A. mackerrasae*	*A. malaysiensis*	Total
mt genome	8 (16 *)	1 (1)	1 (1)	10 (18)
*cox*1	144 (426 ^#^)	5 (5)	11 (23)	160 (454)
*cyt*b	177 (655 ^$^)	0 (0)	76 (76)	253 (731)
*nad*1	0 (130)	0 (0)	0 (0)	0 (130)
SSU	13 (13)	0 (0)	53 (53)	66 (66)
LSU	12 (12)	0 (0)	53 (53)	65 (65)
Total	354 (1252)	6 (6)	194 (206)	554 (1472)

Note: the figures inside and outside brackets represent the number of sequences from GenBank and the number of samples, respectively. SSU: small ribosomal RNAS gene; LSU: large ribosomal RNAS gene. * including 8 genomes from the present study and one genome (KT186242) of *A. malaysiensis* mistaken for *A. cantonensis*. ^#^ including 13 sequences of *A. malaysiensis* mistaken for *A. cantonensis*
^$^ including 64 sequences of *A. malaysiensis* mistaken for *A. cantonensis.*

## Data Availability

The supporting data used in the study are available in the Appendix A.

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
