# Peer review of "The Global Spread Pattern of Rat Lungworm Based on Mitochondrial Genetics"

_pathogens, 2023, doi:10.3390/pathogens12060788_

Round 1

Reviewer 1 Report

Rat lungworm disease caused by Angiostrongylus spp. is an emerging disease of clinical importance. In this study Tian et al., used mitochondrial mapping to consolidate public available data and obtain a global picture of distribution and spread patterns of these pathogens. The published partial sequences of several mitochondrial genes were mapped to whole mitochondrial genomes to uniformly identify their species and clades and map their distribution worldwide. The literature review is particularly valuable and comprehensive.  The manuscript is clearly written and interesting to read. This work contributes to our understanding of the diversity and spread of Angiostrongylus spp. and can serve as a starting point for other studies to come.

Comments

I have only one issue relating to understanding the methodology. As the publication is intended for a wide range of readers, could the authors please explain what is the scientific basis for assuming that each gene fragment used can correctly predict the underlying clade as delineated by whole mitochondrial genomes (is estimating the genetic distances sufficient)?

To that end, would it be interesting to know whether two genes belonging to the same sample (where available) would segregate into the same clade? Would there be a prediction error?

Could the authors please give examples of this methodology used in other studies (references).

I would also consider adding a sentence stating that when incorporated into the individual gene phylogenetic trees, the gene fragments deduced from the whole mitochondrial sequences all remained in their original clades as expected.

Small typing mistake, line 254: “The mt genome (KT186242) previously identified as A. cantonensis was from any known strain of A. cantonensis by over 0.11.

A verb is missing between "was" and "from".

Author Response

Point #1

I have only one issue relating to understanding the methodology. As the publication is intended for a wide range of readers, could the authors please explain what is the scientific basis for assuming that each gene fragment used can correctly predict the underlying clade as delineated by whole mitochondrial genomes (is estimating the genetic distances sufficient)?

Response: Many thanks for this question. We assumed that each gene fragment could predict the clades as inferred by mitochondria genomes based on two previous studies of A. cantonensis. The first was conducted by Chan, A. et al. 2020 [Chan A. et al. Mitochondrial ribosomal genes as novel genetic markers for discrimination of closely related species in the Angiostrongylus cantonensis lineage, Acta Tropica, 2020, 211: 105645]. The 12S and 16S rRNA genes of the same individual specimens were used to re-construct the phylogeny for A. cantonensis. The analysis showed the same phylogenetic relation inferred by the two genes.

The second study was conducted by our team [Lv S. et al. The genetic variation of Angiostrongylus cantonensis in the People’s Republic of China. Infectious Diseases of Poverty, 2017, 6:125]. Different from the first study, we chose protein-encoding genes, i.e., cox1 and nad1, as the markers for a group of the same individual specimens in our study. Since a considerable proportion of singlets in nad1 probably due to sequencing strategy was observed, we did not showed the phylogenetic tree inferred by nad1. However, our analysis indeed showed the consistent phylogenetic relation, although the deep position of haplotypes might be different between the two phylogenies (see the attached figure).

Based on the results of the previous studies, we assumed that the gene fragment can infer the phylogenetic relationship of A. cantonensis samples, and show the consistent result as that inferred by complete mitochondrial genomes at a higher level, e.g., a clade where the p distance among individuals (or connect limit) is less than 1%. Of note, our present study only considered the classification of clades at a p distance less than 1% rather than the exact position of individual samples in trees or sub-trees. Therefore, any sample was assigned to a specific clade according to the criteria (<1%). Otherwise a new clade was defined.

Point #2

To that end, would it be interesting to know whether two genes belonging to the same sample (where available) would segregate into the same clade? Would there be a prediction error?

Response: As mentioned in the point #1, two genes (12S and 16S rRNA genes, or cox1 and nad1) for the same individuals segregate into the same clade. In addition, we added a supplementary file showing the individual gene trees of cox1, cytB, and small rRNA gene reduced from 18 complete mt genomes in the present study. The structure of cox1 gene and cytB gene trees showed the consistent result to the mt genome tree, while the tree of small rRNA gene showed a little bit different in the position of clades. However, the same clades based on the criteria (the p distance among individuals <1%) were identified in all trees. Both previous study and our supplementary file indicate that two genes belonging to the same sample segregated into the same clade.

Point #3

Could the authors please give examples of this methodology used in other studies (references).

Response: We did not found and refer to any previous study where the same methodology was used. However, we believe that our strategy, i.e., categorizing the available mitochondrial genes based on a variety of mitochondrial genomes, is self-standing. First, we hold a variety of complete mitochondrial genomes, including nine which were produced by our team. The selection of the nine mitochondrial genomes from China was based on the previous analysis of population genetics. Most of them (11/15) were from the assumed homeland of A. cantonensis (i.e. Southern China and Southeast Asia) and hence represent the common types in the world. Second, we define a clade as a group of haplotypes with a connection limit < 1% (i.e., p distance is less than 1% between any two haplotypes). Such threshold ensures both the phylogenetic consistence deduced from different genes and the diversity of gene types.

Point #4

I would also consider adding a sentence stating that when incorporated into the individual gene phylogenetic trees, the gene fragments deduced from the whole mitochondrial sequences all remained in their original clades as expected.

Response: We much appreciated this suggestion and has taken it into the revised manuscript (see the text).

Point #5

Small typing mistake, line 254: “The mt genome (KT186242) previously identified as A. cantonensis was from any known strain of A. cantonensis by over 0.11.” A verb is missing between "was" and "from".

Response: We noted this ambiguous description and revised accordingly (see the revised manuscript). Namely, the genetic distance of the mt genome (KT186242), previously identified as A. cantonensis, was over 0.11 from any known strain of A. cantonensis.

Reviewer 2 Report

The study discusses and collets the growing genetic data of A. cantonensis and the unique opportunity it provides to explore the global spread pattern of the parasite. The study's findings reveal new information about the diversity of A. cantonensis in different regions and suggest that systematic research on rat lungworm should be conducted at a global level.

The study also addresses a specific gap in the field by providing new information about the genetic diversity of A. cantonensis in different regions and shedding light on the potential origin of the parasite's spread to new areas. This information can be used to inform public health strategies aimed at preventing and controlling outbreaks of eosinophilic meningitis due to rat lungworm.

It should be noted that this is an in silico paper but it will be very useful to understand the distribution of the nematode by other scientists.

To date, there is no study where the complete sequences of A. cantonenis have been compiled.

The methodology is adequate and the analyses are appropriate to explain the results.

The evidence and arguments presented in the text support the conclusion that systematic research should be conducted at a global level to better understand the spread of the parasite and inform public health strategies.

The tables and figures are consistent with the results.

Author Response

We are grateful for the positive comments.
